# EquiContact: A Hierarchical SE(3) Vision-to-Force Equivariant Policy for Spatially Generalizable Contact-rich Tasks

**Abstract:** This paper presents a framework for learning vision-based robotic policies for contact-rich manipulation tasks that generalize spatially across task configurations. We focus on achieving robust spatial generalization of the policy for the peg-in-hole (PiH) task trained from a small number of demonstrations. We propose EquiContact, a hierarchical policy composed of a high-level vision planner (Diffusion Equivariant Descriptor Field, Diff-EDF) and a novel low-level compliant visuomotor policy (Geometric Compliant ACT, G-CompACT). G-CompACT operates using only localized observations (geometrically consistent error vectors (GCEV), force-torque readings, and wrist-mounted RGB images) and produces actions defined in the end-effector frame. Through these design choices, we show that the entire EquiContact pipeline is $SE(3)$-equivariant, from perception to force control. We also outline three key components for spatially generalizable contact-rich policies: compliance, localized policies, and induced equivariance. Real-world experiments on PiH tasks demonstrate a near-perfect success rate and robust generalization to unseen spatial configurations, validating the proposed framework and principles. The experimental videos are attached as multimedia materials.

**Keywords:** Contact-rich task, Imitation Learning, Compliant Control, Geometric Control, $SE(3)$-Equivariance

## 1 Introduction

Imitation learning has recently shown significant success in expanding the capabilities of machine learning in real-world robotics applications [1, 2, 3, 4, 5, 6, 7]. Similar to the trend seen in large language models (LLMs), there is a growing belief that large-scale data can unlock generalizable, vision-based policies for robotics [8, 1]. However, such policies often lack spatial generalizability and therefore require a large amount of data to learn robust behaviors [9]. An alternative line of recent research focuses on leveraging symmetry—particularly equivariance—to enhance spatial generalizability, thereby improving sample efficiency during training [10, 9].

In this paper, we propose EquiContact, a hierarchical SE(3) vision-to-force equivariant policy for spatially generalizable contact-rich tasks. Our proposed method consists of two main components. The first is a high-level planner consisting of a Diff-EDF model [11], which uses global point cloud data to provide a local reference frame for placing the peg relative to the hole. The second, low-level compliant visuomotor policy is a variant of ACT [6], which we refer to as Geometric Compliant ACT (G-CompACT). This policy handles detailed motion and contact interaction using force-torque feedback and RGB inputs from wrist-mounted cameras. A key design feature of G-CompACT is that it only relies on local information—specifically, the force-torque signal in the end-effector frame, a geometrically consistent error vector (GCEV) [9], and wrist camera inputs. The output of G-CompACT is the desired pose and admittance gains, which are then sent to the geometric admittance controller (GAC) module to execute compliant control. The overall framework of EquiContact is summarized in Fig. 1.

We emphasize that our contribution is on the structural framework rather than the specific choice of algorithms. For example, one could replace the Diff-EDF with other equivariant methods, such as ET-SEED [12], or replace the ACT with the DP or their variants, e.g., Diffusion Transformer (DiT) [13]. The main contributions of this paper are as follows:

Submitted to the 9th Conference on Robot Learning (CoRL 2025). Do not distribute.

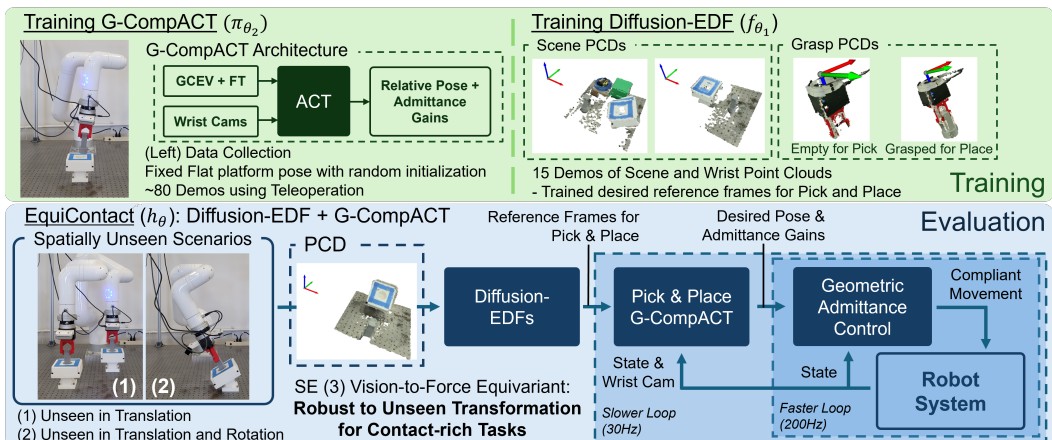

Figure 1: We propose an EquiContact, a hierarchical, provably $SE(3)$ vision-to-force equivariant policy for spatially generalizable contact-rich tasks. The proposed EquiContact consists of G-CompACT and Diffusion-EDF. The G-Compact plays a localized policy over the reference frame provided by the Diffusion-EDF, making our framework generalizable to unseen scenarios during evaluation. The G-CompACT is trained only on the fixed platform pose (left-upper part) but is then deployed to previously unseen platforms, both in translation and rotation (left-lower part), demonstrating $SE(3)$ vision-to-force equivariance and resulting spatial generalization.

1. We propose EquiContact, a hierarchical, provably $SE(3)$-equivariant policy from point clouds and RGB inputs to interaction forces for executing contact-rich tasks.

2. We demonstrate that EquiContact achieves near-perfect success rates and spatial generalizability in real robot experiments involving tight insertions.

3. We identify three key principles for spatially generalizable contact-rich manipulation: Compliance, Localized Policy (Left invariance), and Induced Equivariance. These enable $SE(3)$-equivariant behavior without requiring explicitly equivariant neural networks [10].

From these key principles, we propose a general framework to enhance the spatial generalization and interpretability of vision-based policies, namely, "anchoring localized policy on globally estimated reference frame." We emphasize that our work provides complementary insights to recent trends of robot learning [1, 8, 13, 2] that aim to build generalist policies from a large-scale demo dataset. Our principles offer structural guidelines to improve spatial generalizability through $SE(3)$ equivariance. We note that this work is the workshop version of a previously submitted article, [14].

**Problem Scenario**  Our primary focus is to provide a learning-based solution for a peg-in-hole task, a classic representative of contact-rich, force-based robotic manipulation and assembly tasks, using vision. The robot first needs to pick up the peg and then insert it into the hole, as shown in Fig. S1, relying only on vision and proprioception. We assume only that the peg is upright in the scene. To demonstrate spatial generalizability, we collect expert demonstrations for the G-CompACT in a setting with a fixed, known hole pose. Then, the benchmark and proposed methods trained only with demonstration collected from the fixed hole are evaluated across arbitrarily translated and rotated test scenarios, i.e., spatially out-of-distribution scenarios.

## 2   Solution Approach and Main Result

We now describe our proposed EquiContact policy, which integrates a high-level vision-based planner—Diffusion Equivariant Descriptor Field (Diff-EDF)—with a low-level compliant visuomotor policy—Geometric Compliant control Action Chunking Transformer (G-CompACT). G-CompACT itself consists of a behavior cloning module built on a transformer-based CVAE architecture, followed by a geometric admittance controller (GAC). Because of the page limit, we first focus on the insertion task, which will later be extended to include the picking task to complete the full pipeline implementation. The Diff-EDF and G-CompACT are trained separately but are combined as a whole pipeline during inference time, as shown in Fig. 1.

**Diffusion-Equivariant Descriptor Field (Diff-EDF)**   Diff-EDF [11] is a diffusion-based manipulation policy with a bi-equivariant structure on $SE(3)$ transformations for pick-and-place tasks. Diff-EDF takes two point clouds as input: the scene point cloud $\mathcal{O}^{scene}$ captured by two external RGBD cameras with calibrated extrinsics (Fig. S1), and the gripper point cloud $\mathcal{O}^{grasp}$ in the end-effector frame, with or without a grasped object. Diff-EDF outputs a pose $g_{EDF} \in SE(3)$ representing the estimated target pose of the object of interest, e.g., the pose of the hole, as follows:

$$g_{EDF} = f_{\theta_1}(\mathcal{O}^{scene}, \mathcal{O}^{grasp}) \tag{1}$$

where the neural network $f_{\theta_1}$ implements the mapping. The key property of Diff-EDF is a scene equivariance property, which is written by: $g_l g_{EDF} = f_{\theta_1}(g_l \circ \mathcal{O}^{scene}, \mathcal{O}^{grasp})$ [11]. We refer the readers to Supplementary Material SII.1 for the details of the Diff-EDF.

**Geometric Admittance Control**   We implement the geometric impedance control (GIC) proposed in [15, 16] in the geometric admittance control (GAC) setup [9]. We first note that we follow the notation used in [9]. Let the end-effector pose be denoted as $g \in SE(3)$ in a homogeneous matrix representation, or simply $g = (p, R)$, where $p \in \mathbb{R}^3$ is a position of the end-effector and $R \in SO(3)$ is a rotation matrix of the end-effector. Given the desired end-effector pose $g_d = (p_d, R_d)$, the desired end-effector dynamics for the GAC setup is given as follows:

$$M\dot{V}^b + K_d V^b + f_G = F_e, \ \text{ where } f_G = \begin{bmatrix} f_p \\ f_R \end{bmatrix} = \begin{bmatrix} R^T R_d K_p R_d^T (p - p_d) \\ (K_R R_d^T R - R^T R_d K_R)^\vee \end{bmatrix}, \tag{2}$$

where $M \in \mathbb{R}^{6 \times 6}$ is symmetric positive definite desired inertia matrix, $K_d \in \mathbb{R}^{6 \times 6}$ symmetric positive definite damping matrix, $F_e \in \mathbb{R}^6$ is external wrench applied to the end-effector in end-effector body frame and $V^b \in \mathbb{R}^6$ is a body-frame end-effector velocity. Further, $f_G = f_G(g, g_d, K_p, K_R) \in \mathbb{R}^6$ is a elastic wrench, where $\hat{f}_G \in se^*(3)$, with $K_p, K_R \in \mathbb{R}^{3 \times 3}$ symmetric positive stiffness matrices for the translational and rotational dynamics, respectively, and $(\cdot)^\vee$ denotes the vee-map. The implementation details of GAC are presented in the Supplementary Material SII.2.

**Geometric Compliant control Action Chunking with Transformers (G-CompACT)**   In [9], the authors proposed a recipe for $SE(3)$ equivariant policies, which involves a left invariance property and policy representation in the end-effector body frame. Following this principle and recent developments on relative action representations (similar to [17]), we structure our G-CompACT policy at time instance $k$ to take observations composed of: 1) Geometrically Consistent Error Vector (GCEV) proposed in [9] ($e_G$), 2) FT sensor in the end-effector frame to capture contact behaviors ($F_e$), and 3) RGB images from wrist cameras ($I_w$). The model returns $N$ chunks of action sequences: 1) relative pose from the current end-effector frame ($g_{rel}$), and 2) admittance gains ($\bar{K}_p, \bar{K}_R$). Formally, the G-CompACT can be summarized as

$$\begin{aligned} a(k) &= \pi_{\theta_2}(o(k)), \ \text{ where} \\ a(k) &\triangleq \{(g_{rel}(k+i), \bar{K}_p(k+i), \bar{K}_R(k+i))\}_{i=1}^N, \ o(k) \triangleq (e_G, F_e, I_w)(k), \end{aligned} \tag{3}$$

with $i = 1, \cdots, N$ denoting the index of action chunk. The GCEV $e_G = e_G(g, g_{EDF})$ is defined as

$$e_G(g, g_{EDF}) = \begin{bmatrix} R^T(p - p_{EDF}) \\ (R_{EDF}^T R - R^T R_{EDF})^\vee \end{bmatrix}, \tag{4}$$

where $g$ is current end-effector pose and $g_{EDF}$ is a reference frame obtained from the Diff-EDF. The relative actions $g_{rel}(k+i)$ are projected back to the spatial frame by $\bar{g}_d(k+i) = g(k) \cdot g_{rel}(k+i)$. The temporary signals $(\bar{g}_d, \bar{K}_p, \bar{K}_R)(k+i)$ is then filtered via the temporal aggregation, leading to $(g_d, K_p, K_R)(k)$, which are then provided to the GAC controller (2).

**Main Result: $SE(3)$-Equivariance of EquiContact**   Now we show the main result of EquiContact: $SE(3)$ vision-to-force equivariance. Following the argument of [9], we focus on the elastic wrench $f_G$ in (2) of the GAC equation, since it is the main driving force that pulls the peg into the hole. Let EquiContact be written as $h_\theta$ so that $h_\theta(g, g_{ref}, F_e) \mapsto f_G$, i.e., $h_\theta : SE(3) \times SE(3) \times se^*(3) \to se^*(3)$, where $g_{ref} \in SE(3)$ is the pose representation of the object of interest, e.g., hole assembly (see Fig. S2). As the $g_{ref}$ is unknown, it is observed via $\mathcal{O}^{scene}$ by external RGBD cameras and $I_w$ by wrist cameras. Then, the main proposition that demonstrates $SE(3)$ vision-to-force equivariance is presented as follows:

**Proposition 1.** The EquiContact Policy $h_\theta$ is equivariant if it is described relative to the spatial frame.

Due to space constraints, we will elaborate on the proof and its associated details in the Supplementary Material SII.

# 3 Experimental Result

To validate the effectiveness of our proposed EquiContact framework and its underlying design principles (Compliance, Localized Policy, and Induced Equivariance), we compare EquiContact against three benchmark approaches: ACT with world-frame observation and action representation, executed with and without GAC, and CompACT [18]. We conducted a series of benchmark tests to demonstrate the effectiveness of each component in the proposed approach. The overall results are summarized in Table S1. Note that in this benchmark test, we first focused on the placement task, i.e., the insertion task, and extended the main algorithm to the full pick-and-place task. The full details of the experimental results are shown in the Supplementary Material SIII.

**Demonstration of Compliance** The role of compliance is demonstrated by comparing the same default ACT [6] model running with and without the compliant control in the in the $1^{st}$ and $2^{nd}$ rows of Table S1. Without the compliant control, the ACT model shows significantly lower success rates, which demonstrates that the presence of compliant control is almost a deciding factor between success and failure for contact-rich tasks. To further demonstrate the effectiveness of the task-adaptive gains, we compare the fixed gain case ($2^{nd}$ row) and adaptive gain case (CompACT, $3^{rd}$ row). Although the success rate does not differ, the adaptive admittance gains based on FT feedback consistently produce lower interaction forces in all directions, as shown in Fig. S3.

**Demonstration of Equivariance** Although the CompACT succeeds in insertion tasks in trained scenarios without excessive interaction force, it fails to generalize to spatially unseen configurations (denoted as "OOD" in the $3^{rd}$ row of Table S1). This is expected, as its observation and action representations are defined in the global spatial frame, which neither guarantees nor encourages equivariance. Furthermore, the data is only collected at the fixed platform pose, as shown in Fig. 1. In contrast, the proposed EquiContact achieves perfect success rates on the translationally unseen flat platform, as can be seen in the $4^{th}$ row of Table S1, and achieves a near-perfect success rate even on the tilted platform case.

From these experimental results, we draw the key takeaway: **"anchoring localized policies on globally estimated reference frames"**, which serves as a general framework to enhance spatial generalizability and interpretability of vision-based policies. The detailed elaboration of this takeaway is presented in the Supplementary Material SIV.

# 4 Conclusion

In this work, we introduced EquiContact, a vision-to-force equivariant policy for spatially generalizable contact-rich tasks. By integrating a global reference frame estimator (Diff-EDF) with a fully localized visuomotor servoing policy module (G-CompACT), we demonstrate how compliance, localized policy, and induced equivariance can be unified to enable the peg-in-hole (PiH) task, a representative contact-rich precision task, under spatial perturbations. We proved the $SE(3)$ equivariance of the policy under assumptions on point cloud and image observations and validated its effectiveness through real-world experiments on PiH benchmarks. Compared to benchmark methods, our approach generalizes to unseen platform positions and orientations while maintaining low contact force and near-perfect success rates. Through extensive benchmark studies, we highlighted the effectiveness of the three principles – compliance, localized policy, and induced equivariance – for achieving spatial generalizability in contact-rich manipulation. We conclude that these principles offer a simple yet powerful design guideline for developing spatially generalizable and interpretable robotic policies complementing recent trends in end-to-end visuomotor learning and enabling a structured divide-and-conquer approach.

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

# Supplementary Material

## SI  Related Works

**Visuomotor Servoing Methods** Recently, generative modeling has become mainstream in realizing visuomotor servoing policies. Particularly, there are two dominant methods for visuomotor servoing: Action Chunking with Transformers (ACT) [6] and Diffusion Policy (DP) [5]. ACT utilizes the conditional variational autoencoder (CVAE) method as a generative model, whereas DP employs denoising diffusion as its generative model. ACT and DP have been extended to other approaches, including compliance and force-reactive behaviors [18, 19, 20], as well as structural improvements [21, 13, 22]. Our work is most closely related to CompliantACT (CompACT) [18], which integrates compliant control for visuomotor policies. We have significantly improved CompACT by incorporating a provable $SE(3)$ equivariant structure.

**Equivariant Methods** Equivariant methods aim to leverage the inherent symmetries of robot tasks, typically modeled as group transformations such as $SE(3)$, to improve the sample efficiency of imitation learning. Early approaches, such as Neural Descriptor Fields (NDFs) [23] and Transporter Networks [3], incorporated equivariance to $SE(3)$ and $SO(2)$ transformations, utilizing scene point clouds and top-down views, respectively. Equivariant Descriptor Fields (EDFs) [24] reformulated the NDF approach into a fully end-to-end learning method with $SE(3)$ bi-equivariance property, which is further improved by Diffusion-EDF (Diff-EDF) [11] to improve training and inference performance. Other equivariant models that extract desired keyframes from point clouds include [25]. 3D Equivariant extensions of DP [26, 27, 12] and Flow-Matching [28, 29] policies have been explored in recent literature and combine $SE(3)$ equivariant observation encoders with appropriately modified diffusion/flow-matching processes.

Compared to recent works of [17, 30], we generalize the table-top manipulation to contact-rich tasks while achieving full $SE(3)$ equivariance from vision to force/control level, bridging vision, control, and force interaction under a unified framework. In particular, compared to [30], which represents the target policy on the estimated reference frame, we represent the policy on the end-effector frame. This is because the estimated reference frame can be noisy, but one can always access perfect information on the current end-effector frame via forward kinematics. Furthermore, the reference frame is assumed to be only translated in [30], whereas we assume complete $SE(3)$ transformation of the reference frame. Compared to [17], both [17] and our work propose relying only on the wrist camera for the localized policy. However, while the global information was proposed to be handled via a large field-of-view wrist camera in [17], which is impractical in the real-world implementation, we instead utilized a hierarchical structure to handle global information using an external camera. We also provide more theoretically sound proof for the $SE(3)$ equivariant visuomotor policy, as well as an extension to the control force level. We also did not utilize $SO(2)$ equivariance as in [17] on the wrist camera, as our camera is not mounted vertically on the end-effector. Furthermore, utilizing a group-equivariant neural network in the pipeline tends to increase the computational burden, as most group-equivariant neural networks are not as well-engineered as their non-equivariant counterparts. Therefore, we choose to have a high control frequency for the visuomotor policy rather than utilizing a group-equivariant neural network with a lower control frequency.

## SII  Details of Solution Approach

### SII.1  Details of Diff-EDF

As mentioned in Section 2, the objective of the Diff-EDF is to obtain the pose $g_{EDF} \in SE(3)$, which is the estimated pose of the object of interest $g_{ref}$, as is presented in Fig. S2.

Let $\mathcal{O}^{ref} \subset \mathcal{O}^{scene}$ denote the subset of points corresponding to the object of interest (e.g., the hole structure, see also Fig. S2). The model is designed to satisfy the following left-equivariance property on the point cloud characterization of the target object of interest:

**Assumption 1** (Left-Equivariance of Diff-EDF)**.**

$$g_{EDF} = f_{\theta_1}(\mathcal{O}^{ref}, \mathcal{O}^{grasp}) \implies g_l \cdot g_{EDF} = f_{\theta_1}(g_l \circ \mathcal{O}^{ref}, \mathcal{O}^{grasp}), \qquad (5)$$

To meet this assumption, we randomize the pose of the hole-platform assembly during training, both translationally and rotationally, with visual distractors in the background. This encourages the model to focus on $\mathcal{O}^{ref}$, not the rest of the objects in the scene.

While this property is not strictly enforced (due to the learned nature of $f_{\theta_1}$), Diff-EDF utilizes equivariant backbone architectures and localized attention mechanisms [31], enabling it to generalize from as few as $\sim 10$ demonstrations and have high robustness to left $SE(3)$ transformations.

To train Diff-EDF, the scene and grasp point clouds are collected together with the target reference frames, which represent the desired poses of the end-effector for pick-and-place operations. 15 demonstrations were collected for the Diff-EDF: 12 samples of the flat platform and 3 samples of the tilted platform, both translationally and rotationally randomized. The training process of Diff-EDF follows the procedure in [11].

While the original Diff-EDF supports right-equivariance to handle transformations between the gripper and the grasped object, we did not utilize this feature. In practice, we observed that arbitrary peg transformations relative to the gripper introduced additional challenges. First, grasping the peg based on an arbitrary transformation leads to imprecise grasps, resulting in object slippage during contact. Second, the gripper-object rotation must be estimated continuously in real-time for G-CompACT to work under slippage. Instead, we enforced a consistent grasp orientation and relied only on the left-equivariance property. We refer the readers to [11, 10] for full details of Diff-EDF.

## SII.2 Implementation Details of GAC

Given the desired dynamics (2), the desired end-effector pose command $\tilde{g}_d(k)$ provided to the end-effector controller is calculated in discrete time as

$$
\begin{aligned}
V_d^b(k) &= V^b(k) + T_s \cdot M^{-1}(F_e(k) - f_G(k) - K_d V^b(k)), \\
\tilde{g}_d(k) &= g(k) \cdot \exp(\hat{V}_d^b(k) \cdot T_s),
\end{aligned}
\tag{6}
$$

where $T_s$ is a sampling time and $\hat{(\cdot)}$ denotes a hat-map. For the details of the GIC/GAC, we refer to [15, 9]. The admittance control loop is implemented at a 200Hz rate using ROS2.

## SII.3 Details of G-CompACT

First, we note that in our hardware setup, we utilized two wrist cameras, $I_{w,1}$ and $I_{w,2}$, as shown in Fig. S1, but they are simply denoted as $I_w$. Since the images $I_w$ are fed to the transformer encoder followed by the visual encoder structure, e.g., ResNet, one can further the G-CompACT in (3) as

$$
a(k) = \pi'_{\theta_3}(e_G, F_e, z)(k), \quad \text{where } z(k) = \mu_\phi(I_w).
\tag{7}
$$

where $\mu_\phi$ denotes the visual encoder, and $z$ is the latent variable from the visual encoder. In what follows, we introduce the assumption regarding the left-invariant visual representation.

**Assumption 2** (Left-invariant Visual Features)**.** The features from visual encoder $\mu_\phi$ are left invariant, i.e.,

$$
z(k) = \mu_\phi(g_l \circ I_w) = \mu_\phi(I_w), \quad \forall g_l \in SE(3),
\tag{8}
$$

where we refer to $g_l$ as a left-group action [10]. Originally, $g_l \in SE(3)$ cannot be applied to the image domain. However, with the slight abuse of notation, one can understand the $SE(3)$ action on the image as illustrated in Fig. S2.

The meaning of the visual representation $z$ being left-invariant is that the vision encoder $\mu_\phi$ is trained to focus only on group action invariant features, such as the flat surface surrounding the hole on the platform. The satisfaction of Assumption 2 can be challenging. To satisfy this assumption, we have designed our platform and the surface surrounding the hole assembly so that there are sufficient surface features and the cameras primarily see the surface surrounding the hole, not the lower parts of the platform. This engineering choice was somewhat ad hoc, as there is no guarantee or sufficient inductive bias to encourage the desired behavior. We will demonstrate in later experimental results that this assumption may not apply in certain cases.

The following proposition shows the left-invariance of the G-CompACT method in the end-effector frame.

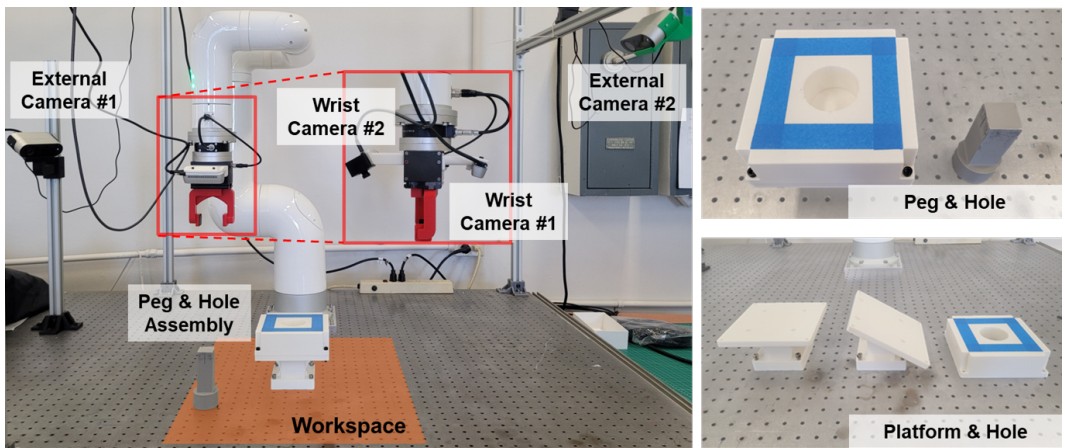

Figure S1: (Left) Overview of the workspace for the peg-in-hole assembly task is presented. 2 external cameras with calibrated extrinsics and 2 wrists cameras are installed. (Right-Top) Peg and hole assembly with 1mm of clearance. (Right-Bottom) Hole part with flat and tilted ($30°$) platforms.

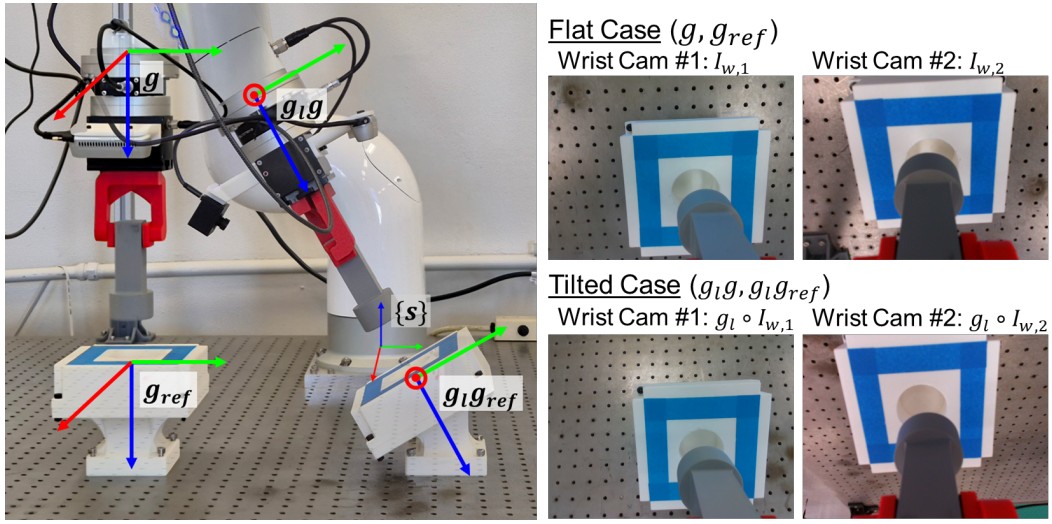

Figure S2: Effects of the left group action $g_l$ to the end-effector pose $g$ and the reference frame $g_{ref}$, and to the wrists cameras $I_{w,1}$ and $I_{w,2}$. As the left group action is applied on the end-effector, the wrist cameras start to see not only the optical table but also the outer backgrounds. $\{s\}$ denotes a spatial frame.

**Proposition 2** (Left-invariance of G-CompACT). Suppose that the Assumption 2 holds. Then,

$$a(k) = \pi_{\theta_2}(g_l \circ o(k)) = \pi_{\theta_2}(o(k)). \tag{9}$$

*Proof.* The left-translated observation signals $o(k)$ reads that:

$$g_l \circ o(k) = (g_l \circ e_G, g_l \circ F_e, g_l \circ I_w). \tag{10}$$

As was shown in Lemma 1 of [9], the GCEV $e_G$ is left invariant as

$$g_l \circ e_G(g, g_{EDF}) = e_G(g_l g, g_l g_{EDF}) = e_G(g, g_{EDF}).$$

The force-torque sensor values are left-invariant because they are already defined with respect to the end-effector frame [9], and the visual representation vectors satisfy left invariance due to Assumption 1. Combining all these properties, it follows that

$$a(k) = \pi_{\theta_2}(g_l \circ o(k)) = \pi'_{\theta_3}(e_G, F_e, z) = \pi_{\theta_2}(o(k)), \tag{11}$$

which shows the left invariance of the G-CompACT policy on the end-effector frame. □

In what follows, we demonstrate that the pose signal produced by G-CompACT is equivariant when described in the spatial frame.

**Corollary 1** ($SE(3)$ Equivariance of G-CompACT). Let the output of G-CompACT in the spatial frame be $\bar{g}_d$ via (without considering the outputs related to gains)

$$\bar{g}_d = g \cdot \pi_{\theta_2}(e_G(g, g_{EDF}), F_e, I_w). \tag{12}$$

Then,

$$\begin{aligned} g_l \bar{g}_d &= g_l g \cdot \pi_{\theta_2}(g_l \circ e_G(g, g_{EDF}), g_l \circ F_e, g_l \circ I_w) \\ &= g_l g \cdot \pi_{\theta_2}(e_G(g, g_{EDF}), F_e, I_w). \end{aligned} \tag{13}$$

Since the temporary desired pose on the spatial frame signal, $\bar{g}_d$, is $SE(3)$ equivariant, the temporally ensembled signal $g_d$ is also equivariant. Therefore, by combining the G-CompACT with a spatial frame representation and temporal ensemble, we will now define the final form of G-CompACT as $\hat{\pi}_{\theta_2}$, which is given by:

$$(g_d, K_p, K_R) = \hat{\pi}_{\theta_2}(e_G(g, g_{EDF}), F_e, I_w). \tag{14}$$

and satisfies the following equivariance property:

$$(g_l g_d, K_p, K_R)(k) = \hat{\pi}_{\theta_2}(g_l \circ o(k)). \tag{15}$$

To train a G-CompACT, we collect the expert demonstration using teleoperation on the fixed pose of the platform. During data collection, the expert teleoperator monitors the task's progress, makes real-time movement commands via a SpaceMouse, and adjusts the admittance gains using keyboard input to switch between the following predefined gain modes: low-gain mode, high-gain mode, insertion mode, and contact mode. The low/high gain mode has low/high gains in all directions, the insertion mode has high gains in the $z$ direction of the end-effector frame and low gains elsewhere. Finally, the contact mode has low gains in the $z$ direction and high gains elsewhere. We collected 86 demonstrations to train a near-perfect policy.

Since we know the fixed pose of the platform a priori, e.g., a ground-truth reference frame, the GCEV vector can be calculated for the training process. Nevertheless, the reference frame needs to be estimated via Diff-EDF (as $g_{EDF}$) during the inference stage, which may have non-negligible errors. To handle this issue, we have added noise to the reference frame $g_{ref}$ to calculate $e_G$ during dataset preprocessing. This provides the model with an inductive bias to primarily rely on $e_G$ values for rough alignment and rely on vision feedback for fine-grained motion. The rest of the training follows the standard imitation learning pipeline.

### SII.4 EquiContact

The proposed EquiContact method comprises the high-level Diff-EDF, which serves as a high-level vision planner that provides the reference frame to be fed to GCEV, and the low-level G-CompACT, which handles fine-grained movement and contact interaction during insertion using real-time vision and force feedback. The overall pipeline of the EquiContact is presented in Fig. 1. The $SE(3)$ vision-to-force equivariance property of EquiContact is proposed in Proposition 1; we now present the proof here.

*Proof.* Suppose that the Assumption 1 and 2 hold. Let the object of interest, e.g., a peg for the picking task and a hole for the placing task, be observed by $\mathcal{O}^{ref}$, $I_w$ with its pose given by $g_{ref}$, so that the left-translated $g_l \cdot g_{ref}$ is observed by $g_l \circ \mathcal{O}^{ref}$ from the point cloud, and $g_l \circ I_w$ by the left-translated end-effector attached wrist camera as described in Fig. S2. First, notice that $h_\theta$ can be fully written as

$$\begin{aligned} h_\theta(g, g_{ref}, F_e) &= f_G(g, \hat{\pi}_{\theta_2}(e_G(g, f_{\theta_1}(\mathcal{O}^{ref})), F_e, I_w)) \\ &= f_G(g, \underbrace{\hat{\pi}_{\theta_2}(e_G(g, g_{EDF}), F_e, I_w)}_{=(g_d, K_p, K_R)}). \end{aligned} \tag{16}$$

Then, when both $g$ and $g_{ref}$ undergoes a left transformation $g_l$, from Assumption 1 and Corollary 1, the following holds:

$$
\begin{aligned}
& h_\theta(g_l g, g_l g_{ref}, g_l \circ F_e) \\
&= f_G(g_l g, \hat{\pi}_{\theta_2}(e_G(g_l g, f_{\theta_1}(g_l \circ \mathcal{O}^{ref})), g_l \circ F_e, g_l \circ I_w)) \\
&= f_G(g_l g, \hat{\pi}_{\theta_2}(e_G(g_l g, g_l g_{EDF}), g_l \circ F_e, g_l \circ I_w)) \\
&= f_G(g_l g, g_l g_d, K_p, K_R) = f_G(g, g_d, K_p, K_R) \\
&= h_\theta(g, g_{ref}, F_e).
\end{aligned}
\tag{17}
$$

We note that the second-last equation ($SE(3)$ left-invariance of the elastic wrench) comes from Lemma 1 of [9]. Finally, from the result of Proposition 2 of [9], it follows that

$$
h_\theta^s(g_l g, g_l g_{ref}, g_l \circ F_e) = \mathrm{Ad}_{g_l^{-1}}^T h_\theta^s(g, g_{ref}, F_e),
\tag{18}
$$

where Ad is an Adjoint operator, $\mathrm{Ad}_{g_l^{-1}}^T$ is a representation for $se^*(3)$ (wrench) domain for the group action $g_l$ [9], and superscript $s$ denotes a vector represented in the spatial frame. $\square$

**Extensions to Pick Tasks** So far, we have described our method in terms of the insertion (placement) task. The proposed method can be extended to pick tasks in the same manner. The Diff-EDF can be utilized to obtain the pick reference frame, which is used for $e_G$ for the picking G-CompACT. The picking G-CompACT is trained in such a way that the manipulator grasps a peg in a fixed, aligned pose, which helps EquiContact bypass the right-equivariance issue. For G-CompACT, the FT sensor values are not utilized as one of its observations, and it does not output the admittance gains; instead, it uses fixed gains.

## SIII  Additional Experimental Results

Here, we present additional experimental results for EquiContact: a result of the full pick-and-place pipeline and the failure cases due to dissatisfaction of Assumption 2.

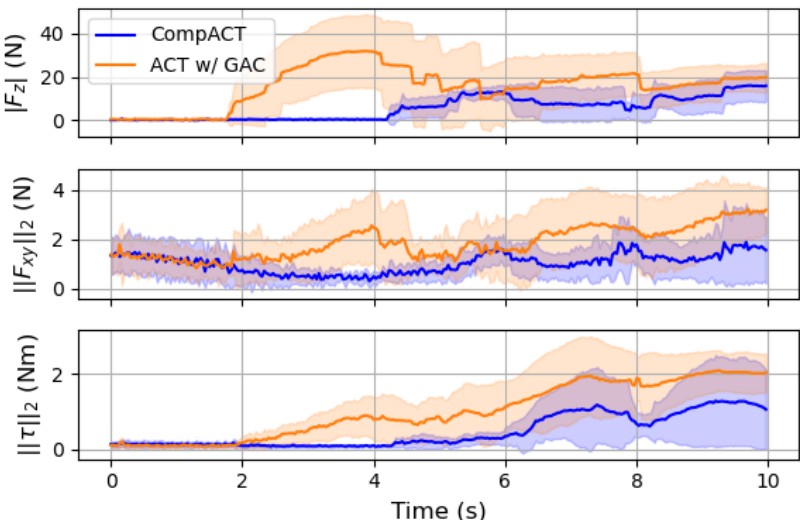

Figure S3: Force profiles of CompACT and ACT with GAC (fixed gains) during insertion tasks are presented. The CompACT with force-torque sensor inputs and output gains shows lower interaction force in all directions.

**Full Pick-and-place Pipeline** The result of EquiContact for the full pick-and-place task is summarized in Table S2. The EquiContact also demonstrates a near-perfect success rate in the full pick-and-place pipeline for peg-in-hole tasks. However, as the whole task is formulated in sequential stages, the error in the previous stage tends to propagate to the following stages, leading to slightly increased failure cases.

Table S1: Success rates of the insertion policies in real-world experiments for the proposed and benchmark approaches. "In-Dist." denotes in-distribution data and "OOD" denotes out-of-distribution data. In addition, "Flat" and "Titled" in the Test Scenario refer to testing with the flat and tilted platforms, respectively. For the In-Dist. (in distribution) scenario, the initial pose of the end-effector is randomized around the flat platform.

| Methods | Observation | Action | Test Scenario | Success Rate |
|---|---|---|---|---|
| ACT w/o GAC | `World Pose` | `World Pose` | Flat (In-Dist.) | 2 / 10 |
| ACT w/ GAC | `World Pose` | `World Pose` | Flat (In-Dist.) | 20 / 20 |
| CompACT | `World Pose, FT` | `World Pose, Gains` | Flat (In-Dist.) | 20 / 20 |
| | | | Flat (OOD) | 0 / 10 |
| **EquiContact (Place, Ours)** | `GCEV, FT` | `Relative Pose, Gains` | Flat (OOD) | 20 / 20 |
| | | | Tilted ($30°$, OOD) | 19 / 20 |

Table S2: Success Rates of the proposed EquiContact for a full pipeline of pick and place.

| Test Scenario | Success Rate | Failure Cases |
|---|---|---|
| Flat Platform (OOD) | 20 / 20 | N/A |
| Tilted Platform ($30°$, OOD) | 18 / 20 | 1 Pick, 1 Place |

Table S3: Success Rates of the G-CompACT (insertion) trained with the base dataset and augmented dataset for the flat platform under the presence of visual distractor scenario and tilted platform with large angle cases. The evaluation is conducted with the ground-truth reference frames.

| Test Scenario | Base Dataset | Augmented Dataset |
|---|---|---|
| W/ Visual Distractor | 4 / 10 | 9 / 10 |
| Tilted Platform ($45°$) | 5 / 10 | 9 / 10 |

**Failure Cases and Performance Recovery via Data Augmentation**  Although the EquiContact framework is vision-to-force equivariant in theory, this guarantee holds only under Assumptions 1 and 2. In particular, Assumption 2, which requires left-invariance of the visual features, is more challenging to enforce in practice, as our approach does not explicitly encode this property through loss functions or architectural inductive bias. As a result, the proposed G-CompACT algorithm shows degraded performance in scenarios with visual distractors or on a more severely tilted platform ($45°$) – see Table S3 ("Base Dataset" column). Note that the larger tilting angle results in more out-of-distribution images, as the wrist cameras begin to capture more unseen background scenes, as illustrated in Fig. S2. To accommodate this, we augment the base dataset with 20 demos collected using the visual distractor on a fixed, flat platform and 20 demos collected from a $30°$ angle of the tilted platform. After training with the augmented dataset, the success rates return to normal levels, as shown in the "Augmented Dataset" column of Table S3, demonstrating the performance recovery achieved through data augmentation.

## SIV    Anchoring a Localized Policy on the Reference Frame for Equivariance

The G-CompACT policy $\pi_{\theta_2}$ operates exclusively on localized inputs: GCEV, FT values in the end-effector frame, and images from wrist cameras. Its outputs, relative poses, and admittance gains are likewise defined in the local end-effector frame. The localized policy is anchored to a reference frame generated by the Diff-EDF planner. This architectural design induces $SE(3)$-equivariance and resulting spatial generalization. Under this principle, one may hypothesize the reason for the spatial generalizability of recent vision-language-action models, which utilize both wrist and external cameras. The wrist camera provides a localized policy, anchored in the reference frame obtained from the external cameras and proprioceptive information. In fact, learning a reference frame from proprioceptive information (e.g., joint position) and the external camera is subtle because of its black-box nature. In contrast, our method, which utilizes a point cloud-based reference frame

approach, is more interpretable. Thus, we propose **"anchoring localized policies on globally esti-mated reference frames"** as a general framework for the divide-and-conquer philosophy to enhance spatial generalization and interpretability. In our case, this structure also enables provable $SE(3)$ equivariance via differential geometric design.

