# OpenReview forum: "EquiContact: A Hierarchical SE(3) Vision-to-Force Equivariant Policy for Spatially Generalizable Contact-rich Tasks"
_robot-learning.org/CoRL/2025/Workshop/Dexterous_Manipulation — CoRL 2025 Workshop Dexterous Manipulation Spotlight_

### Official Review · Reviewer_yuWE · 2025-09-09
**Interesting work to improve sample-efficiency**

**Rating:** 7
**Confidence:** 4

**Review:**

This work proposes EquiContact, a pipeline for vision-based robotic manipulation. The pipeline consists of two parts: Diff-EDF and Geometric Compliant ACT (G-CompACT). These two models are largely based on prior work, but the authors add an equivariance property to CompACT. Their proposed pipeline is modular; Diff-EDF and G-CompACT are trained separately and interact via explicit representations in contrast to end-to-end learned approaches. With this pipeline, the authors focus on achieving high success rates for a challenging peg insertion task with few samples. To achieve high success rate in a sample-efficient manner, the authors rely on equivarance throughout their entire pipeline, which aids with spatial generalization given a low number of demonstrations. Furthermore, the authors show that admittance control is crucial for high success rate in this task.

Strengths:
- Using equivariance to improve sample-efficiency is a thoughtful approach that seems to work quite well
- Very detailed description of methodology
- High success rates in a quite challenging task

Weaknesses:
- The design choice of training two separate equivariant models instead of one equivariant end-to-end model should be studied - the tradeoffs here are not immediately clear and this paper would benefit from more end-to-end baselines (i.e. equivariant diffusion policy)
- More tasks that highlight the strengths of this framework would be nice

Also a small remark - the disclaimer that the paper was submitted elsewhere includes an arXiv reference, which technically violates the anonymity of the review process...

---

### Official Review · Reviewer_iFuE · 2025-09-10

**Rating:** 7
**Confidence:** 4

**Review:**

This paper proposes EquiContact, a hierarchical policy framework for contact-rich manipulation tasks (with peg-in-hole as the primary benchmark). The method combines:
1. A high-level planner (Diff-EDF) to establish a reference frame using point clouds.
2. A low-level visuomotor controller (G-CompACT) that operates with force-torque sensing, wrist-mounted RGB images, and geometrically consistent error vectors.
The key novelty is ensuring that the entire perception-to-control pipeline is SE(3)-equivariant, enabling spatial generalization from limited demonstrations. Experiments show near-perfect success rates in both in-distribution and out-of-distribution scenarios (translations and rotations of the platform).

Strengths
1. This paper addresses the challenge of spatial generalization in contact-rich tasks, where both data efficiency and robustness are critical.
2. It identifies three key design principles—compliance, localized policies, and induced equivariance—integrates them coherently, and provides theoretical grounding for SE(3) equivariance of the policy, with supporting proofs in the supplementary material.
3. Real-robot experiments on peg-in-hole tasks with both flat and tilted platforms demonstrate that the proposed method achieves near-perfect success rates in unseen configurations, outperforming ACT and CompACT baselines. Additional analyses of compliance (with vs. without GAC) and dataset augmentation further demonstrate robustness.

Weaknesses and Limitations
1. Evaluation is limited to peg-in-hole assembly. While this is a classic benchmark, the generality to other contact-rich tasks (e.g., pushing, surface following, multi-object assembly) remains unclear.
2. While the method claims to work with a small number of demonstrations (~15–20 for Diff-EDF, ~80 for G-CompACT), it is unclear how this scales with more complex tasks or environments.
3. Baselines are mostly ACT/CompACT variations. Missing comparisons with recent diffusion-based visuomotor policies, which might also generalize well.
4. The picking phase is addressed only briefly and without force sensing or adaptive gains, reducing the strength of the “full pipeline” contribution.

---

### Decision · Program_Chairs · 2025-09-18

Accept (Spotlight)